# Antibody and T Cell Responses against SARS-CoV-2 Elicited by the Third Dose of BBIBP-CorV (Sinopharm) and BNT162b2 (Pfizer-BioNTech) Vaccines Using a Homologous or Heterologous Booster Vaccination Strategy

**DOI:** 10.3390/vaccines10040539

**Published:** 2022-03-30

**Authors:** Zsolt Matula, Márton Gönczi, Gabriella Bekő, Béla Kádár, Éva Ajzner, Ferenc Uher, István Vályi-Nagy

**Affiliations:** 1Laboratory for Experimental Cell Therapy, Central Hospital of Southern Pest, National Institute of Hematology and Infectious Diseases, 1097 Budapest, Hungary; uher.ferenc@dpckorhaz.hu; 2Central Laboratory of Central Hospital of Southern Pest, National Institute of Hematology and Infectious Diseases, 1097 Budapest, Hungary; gonczi.marton@dpckorhaz.hu (M.G.); beko.gabriella@dpckorhaz.hu (G.B.); kadar.bela@dpckorhaz.hu (B.K.); 3Department of Hematology and Stem Cell Transplantation, Central Hospital of Southern Pest, National Institute of Hematology and Infectious Diseases, 1097 Budapest, Hungary; ajzner.eva@dpckorhaz.hu (É.A.); valyi-nagy.istvan@dpckorhaz.hu (I.V.-N.)

**Keywords:** adaptive immunity, anti-SARS-CoV-2 antibodies, IFNγ-producing T cells, IFNγ ELISpot assay, heterologous prime-boost vaccination

## Abstract

In the present study, antibody and T cell-mediated immune responses elicited by BBIBP-CorV and BNT162b2 vaccines were compared 6 months after the two-dose immunization of healthy individuals. Additionally, antibody and T cell responses after the third dose of BBIBP-CorV or BNT162b2 were compared using a homologous or heterologous vaccination strategy. The third dose was consistently administered 6 months after the second dose. Six months following the two-dose vaccination, the cumulative IFNγ-positive T cell response was almost identical in participants immunized with either two doses of BNT162b2 or BBIBP-CorV vaccines; however, significant differences were revealed regarding humoral immunity: the two-dose BNT162b2 vaccine maintained a significantly higher antireceptor-binding domain (RBD) IgG, anti-spike (S1/S2) IgG, and IgA antibody levels. The BNT162b2 + BNT162b2 + BBIBP-CorV vaccine series elicited significantly lower anti-RBD IgG and anti-S1/S2 IgG levels than three doses of BNT162b2, while the anti-S IgA level was equally negligible in both groups. Importantly, the cumulative IFNγ-positive T cell response was highly similar in both groups. Surprisingly, the BBIBP-CorV + BBIBP-CorV + BNT162b2 vaccination series provided a much higher cumulative IFNγ-positive T cell response than that elicited by three doses of BNT162b2; moreover, the levels of anti-RBD IgG and anti-S IgA were almost identical. Only the mean anti-S1/S2 IgG levels were higher after receiving three mRNA vaccines. Based on these data, we can conclude that administering a third dose of BNT162b2 after two doses of BBIBP-CorV is an effective strategy to significantly enhance both humoral and T cell-mediated immune response, and its effectiveness is comparable to that of three BNT162b2 vaccines.

## 1. Introduction

Inactivated virus vaccines against SARS-CoV-2 have played an important role in promoting worldwide vaccine coverage in the past year [1]. According to the latest globally summarized data, the Chinese BBIBP-CorV (Sinopharm, Beijing, China) and CoronaVac (Sinovac, Beijing, China) vaccines account for nearly half of all COVID-19 vaccine doses or nearly 5 billion of the more than 11 billion COVID-19 vaccines administered worldwide to date. Moreover, this number is further raised by the more than 200 million doses of other inactivated virus vaccines, such as Covaxin from India, COVIran Barekat from Iran, and QazVac from Kazakhstan [2]. In terms of vaccine efficacy, the two most widely used inactivated virus vaccines globally, BBIBP-CorV and CoronaVac, are 78.1% and 50.7% effective against PCR-confirmed symptomatic COVID-19, respectively [3,4,5]. Furthermore, the inactivated virus vaccines elicit robust humoral responses against SARS-CoV-2 infection in participants aged 3–17 years, showing good safety and tolerability profiles for this age group [6].

Although these widely used vaccines remain crucial for preventing hospitalization and death worldwide, as they can still serve a valuable immune-priming potential for unvaccinated individuals, it has recently been revealed that they provide little or no protection against the rapidly spreading B.1.1.529 (Omicron) variant. A rising number of papers reveal that many people who receive two doses of an inactivated vaccine fail to produce enough neutralizing antibodies against the spike protein to prevent Omicron transmission [7,8]; however, this is also the case for mRNA and recombinant adenovirus vaccines after two doses [9,10]. Moreover, after the third dose of an inactivated virus vaccine, the level of anti-spike neutralizing antibody highly improves, but is probably still not high enough against Omicron [11,12]. Additionally, some recent research revealed that a third dose of another type of vaccine, such as mRNA (BNT162b2) or adenoviral vectored vaccines (ChAdOx1, Oxford-AstraZeneca; Ad26.COV2.S, Janssen, New Brunswick, NJ, USA), after a two-dose inactivated virus vaccine regimen (CoronaVac) provides much better protection against the Omicron variant [13,14,15,16]. Unfortunately, these data only provide information about antibody responses, while the magnitude of T cell responses elicited by homologous or heterologous boosters is unclear.

To supplement this incomplete knowledge, we determined the humoral and T cell-mediated immune responses against SARS-CoV-2 elicited by the BNT162b2 (Pfizer-BioNTech, Mainz, Germany) vaccine following two doses of the BBIBP-CorV vaccine and compared them to a homologous vaccination strategy using three doses of the BNT162b2 vaccine. Moreover, the effects on the adaptive immune response elicited by a third, heterologous vaccine combination were also analyzed—where participants received an inactivated virus vaccine (BBIBP-CorV) after two doses of an mRNA vaccine (BNT162b2). The third dose of vaccine was administered 6 months after the second dose in all vaccine combinations. Specimens from a total of 52 healthy individuals were investigated regarding the humoral and T cell-mediated immune responses using serological assays and T cell ELISpot assay. We found that 6 months after the two-dose BNT162b2 or BBIBP-CorV vaccination, both the T cell-mediated and humoral immune responses were significantly reduced compared to those values measured two weeks after the second vaccination, the results of which were reported in our previous study [17]. Although the cumulative T cell response was almost the same, significantly higher levels of anti-S1/S2 IgG, anti-RBD IgG, and anti-S IgA were measured six months after the administration of the two-dose BNT162b2 compared to six months after the administration of the two-dose BBIBP-CorV vaccine regimen. A booster dose of the BNT162b2 vaccine 6 months after the administration of the two-dose BBIBP-CorV vaccine regimen induces an even stronger T cell response than three doses of BNT162b2 and also provides a massive humoral response, the magnitude of which is remarkably similar to that elicited by three doses of BNT162b2. In contrast, for the BNT162b2 + BNT162b2 + BBIBP-CorV vaccine combination, much lower anti-spike (anti-S1/S2) IgG, anti-spike (anti-S) IgA, and neutralizing antireceptor-binding domain (anti-RBD) IgG antibody levels, but the same robust T cell response was measured as after three doses of BNT162b2. These findings may help scientists and clinicians reconsider the role of inactivated virus vaccines in the global fight against COVID-19.

## 2. Materials and Methods

### 2.1. Determination of SARS-CoV-2-Specific T Cell Response

SARS-CoV-2-specific T cell immunity was determined using a T-SPOT Discovery SARS-CoV-2 kit (Oxford Immunotec Ltd., Abingdon, UK) that quantifies the IFNγ-producing T cells in response to SARS-CoV-2-specific viral peptides. Peripheral blood samples were collected in sodium citrate vacutainer tubes (BD Biosciences, San Jose, CA, USA), and peripheral blood mononuclear cells (PBMCs) were isolated by density gradient centrifugation using the Leucosep Kit (Oxford Immunotec) according to the manufacturer’s instruction. A total of 250,000 viable PBMCs were plated into each well of the T-SPOT Discovery SARS-CoV-2 kit, which is composed of five different but overlapping peptide pools to cover protein sequences of five different SARS-CoV-2 antigens including spike (S1 and S2), nucleocapsid (N), membrane (M), and envelope (E). Peptides showing high sequence homology to endemic coronaviruses are removed from the peptide pools by the manufacturer. The cumulative spot-forming units (SFU) per 2.5 × 10^5^ PBMC of individuals were calculated as the total number of T-spots for S1, S2, N, M, and E antigens minus the background for each antigen.

### 2.2. Determination of SARS-CoV-2-Specific Antibody Response

Peripheral blood samples were collected in serum vacutainer tubes (BD Biosciences) and centrifuged for 10 min at 3500 g at room temperature. Serum samples were stored at −80 °C. The levels of anti-SARS-CoV-2 antibodies were evaluated using commercially available test systems. A SARS-CoV-2 Surrogate Virus Neutralization Test (sVNT) Kit (GenScript Biotech B.V., Leiden, The Netherlands) was used to detect IgG levels against SARS-CoV-2 RBD. A LIAISON SARS-CoV-2 S1/S2 IgG test (DIASORIN S.P.A., Saluggia, Italy) was utilized to detect anti-S1/S2 IgG antibodies, and SARS-CoV-2 anti-S IgA assay (EUROIMMUN Medizinische Labordiagnostika AG, Lübeck, Germany) was used to evaluate IgA levels against SARS-CoV-2 spike protein.

### 2.3. SARS-CoV-2-Specific Antibody and T Cell Responses 6 Months after Two-Dose BBIBP-CorV or BNT162b2 Vaccine Regimen

We determined the SARS-CoV-2-specific antibody and T cell responses 6 months after administration of the two-dose vaccine (BBIBP-CorV or BNT162b2) among 52 healthy adult volunteers. Antibody response was determined by measuring the levels of anti-RBD IgG, anti-S1/S2 IgG, and IgA antibodies, while T cell response was determined by quantifying the IFN-γ producing T cells after exposure to S1 and S2, N, M, and E antigen peptides, as described previously. Altogether, 24 participants received two doses of the BBIBP-CorV (Sinopharm’s Beijing Institute of Biological Products, Beijing, China), and 25 received two doses of the BNT162b2 (Pfizer-BioNTech, Pfizer Inc., New York, NY, USA) vaccine. Exclusion criteria were fever, cough, and diarrhea one week before assessment. Additionally, all participants tested negative for active infection on the day of blood collection by PCR assay. These specimens were obtained between 19 September 2021 and 30 September 2021. The median age and age range of the participants was 51 (19–64).

### 2.4. SARS-CoV-2-Specific Antibody and T Cell Responses after Third BNT162b2 Vaccine following a Two-Dose BNT162b2 or BBIBP-CorV Vaccine Regimen, or after Third BBIBP-CorV Vaccine following a Two-Dose BNT162b2 Vaccine Regimen

The SARS-CoV-2-specific antibody and T cell responses were analyzed after the third dose of the BNT162b2 or BBIBP-CorV vaccine. A total of three groups were formed based on the vaccine combinations, of which the first group received a homologous vaccine, while the second and the third received a heterologous vaccine regimen. The first group was inoculated by the BNT162b2 vaccine following the two-dose BNT162b2 (*n* = 8); the second group received the BBIBP-CorV vaccine following the two-dose BNT162b2 (*n* = 13), while the third group was inoculated by the BNT162b2 vaccine following the two-dose BBIBP-CorV vaccine (*n* = 10). Six participants were immunized with two doses of the BNT162b2 vaccine and boosted with BNT162b2 (*n* = 3) or BBIBP-CorV (*n* = 3) vaccines; however, since between the second and third vaccinations they were infected with SARS-CoV-2, they had hybrid immunity. The vaccination schedule, groups of participants, and sample collection of the present study are shown in Figure 1. These specimens were obtained between 27 September 2021 and 19 October 2021. The median age was 52, and the age range of the participants was (29–64).

### 2.5. Statistical Analysis

Statistical analyses were performed by applying the Wilcoxon signed-rank test or Student’s *t*-test as appropriate, and *p*-values of <0.05 were considered to be statistically significant.

## 3. Results

### 3.1. Antibody and T Cell Responses against SARS-CoV-2 6 Months after Immunization with Two-Dose BNT162b2 or BBIBP-CorV Vaccine

Blood samples were analyzed 6 months following the second dose of the BNT162b2 (Pfizer-BioNTech) or BBIBP-CorV (Sinopharm) vaccines regarding the antibody and T cell-mediated immune response (Figure 1A). According to the results of anti-SARS-CoV-2 ELISpot assays, the data of 6 participants were evaluated separately within the BNT162b2 cohort: study participants with ≤8 SFU/2.5 × 10^5^ PBMC for N and M peptide pools were considered virus-naive individuals, while the 6 participants with ≥8 SFU/2.5 × 10^5^ PBMC for N and M antigens were considered as individuals who probably experienced mild or asymptomatic SARS-CoV-2 infection after receiving the second dose of the vaccine. Significant differences were observed within the BNT162b2 cohort between the virus-naive and virus-experienced individuals in terms of N and M antigen response in ELISpot assays, and additionally, the cumulative T cell response was more than twice as high in the virus-experienced group than in the virus-naive group (mean 119.2 SFU vs. 47.9 SFU) (Appendix A). In the case of virus-naive participants within the BNT162b2 cohort, the cumulative T cell response was almost identical (mean 47.9 SFU) to that in the BBIBP-CorV cohort (mean 49.9 SFU) six months after immunization (Figure 2A). However, in the BBIBP-CorV cohort, it was not possible to determine how many participants were infected with the SARS-CoV-2 virus after the two vaccinations since, in contrast to the mRNA vaccine, the inactivated virus vaccine can induce an immune response not only against S but also against N, M, and E antigens.

In contrast to T cell immunity, significant differences in antibody response were observed between the two cohorts. Participants who previously received the two mRNA vaccines (BNT162b2) had significantly higher levels of anti-S1/S2 IgG (mean 95.3 AU/mL) and anti-RBD antibodies (mean 65.8 neutr (%)) after 6 months than those who received two inactivated virus vaccines (BBIBP-CorV) (mean 20.8 AU/mL and mean 15.3 neutr (%), respectively) (Figure 2B,C). The anti-S IgA antibody levels were equally negligible in both cohorts, with no significant difference between them (mean 2.1 S/CO vs. 0.5 S/CO) (Figure 2D). Finally, slightly higher antibody levels were measured in those participants who were likely to be infected than in virus-naive participants (Appendix A). After 6 months, both the T cell-mediated and antibody responses were significantly reduced in both cohorts compared to those measured two weeks after the second vaccination of the same participants. The percentages of these reductions are shown in Figure 2 below the *X*-axis for each parameter.

### 3.2. SARS-CoV-2-Specific Antibody and T Cell Responses after the Third (Booster) Dose of BNT162b2 or BBIBP-CorV Vaccines

The adaptive immune response elicited by the third dose of the BNT162b2 or BBIBP-CorV vaccine was analyzed 14 days after the third vaccination. All participants were 6 months after their initial two-dose BNT162b2 or BBIBP-CorV vaccination series at the time of receiving the third dose (Figure 1A). In total, based on their vaccine combinations, three groups were tested to compare the homologous and heterologous vaccination strategies (Figure 1B). The first group was immunized by the BNT162b2 vaccine following the two-dose BNT162b2 (*n* = 8); the second group received the BBIBP-CorV vaccine following the two-dose BNT162b2 (*n* = 13), while the third group was boosted by the BNT162b2 vaccine following the two-dose BBIBP-CorV vaccine (*n* = 10). Regarding the anti-SARS-CoV-2 T cell response, highly similar results were obtained in the case of virus-naive individuals of group 1 and 2 two weeks after the third dose, where participants of group 1 had a mean value of 121 SFU/250,000 PBMCs, while participants of group 2 had a mean value of 117 SFU/250,000 PBMCs (Figure 3A). In group 3, where participants were boosted by BNT162b2 following the two-dose BBIBP-CorV vaccine, the cumulative IFNγ-positive T cell response was surprisingly much higher (Figure 3A, mean 172.9 SFU/250,000 PBMC) than in groups 1 and 2, where participants were immunized with two doses of the BNT162b2 and boosted with either the BNT162b2 or BBIBP-CorV vaccine without SARS-CoV-2 infection.

A small subset of participants was immunized with two doses of the BNT162b2 vaccine and boosted with the BNT162b2 (*n* = 3) or BBIBP-CorV (*n* = 3) vaccines, but between the second and third vaccinations, they were probably all infected with the SARS-CoV-2 virus. The mean SFU of these participants with hybrid immunity was 209.8 after the third dose, but these individuals evidently started from a higher baseline due to the infection following the two-dose vaccination (Appendix A).

In contrast to T cell immunity, significant differences in antibody response were observed between the three groups. Virus-naive participants in group 1 who were previously immunized with two BNT162b2 vaccines and boosted with a third (mRNA) vaccine had significantly higher anti-S1/S2 IgG (mean 8272.2 AU/mL), anti-RBD IgG (mean 98.3 neutr (%)), as well as anti-S IgA antibody levels (mean 11.2 S/CO) than the virus-naive participants in group 2 who received the BBIBP-CorV vaccine following the two-dose BNT162b2 vaccine regimen (mean 1195.8 AU/mL, 79 neutr (%), 5.7 S/CO, respectively) (Figure 3B–D). Group 3 participants who had previously been immunized with two doses of an inactivated virus vaccine (BBIBP-CorV) and then boosted with an mRNA (BNT162b2) vaccine had similar anti-RBD IgG antibody (mean 99.2 neutr (%)) and anti-S IgA antibody levels (mean 12.0 S/CO), but lower levels of anti-S1/S2 IgG antibody (mean 3434.6 AU/mL) (Figure 3B–D) compared to virus-naive participants of group 1 who received three doses of the BNT162b2 vaccine. However, the difference between the anti-S1/S2 IgG levels was not statistically significant due to the high interindividual differences in both groups.

Although the six participants with hybrid immunity (two doses of mRNA vaccine + infection) started from higher antibody levels due to infection, after the booster dose (BNT162b2), only their anti-S IgA levels were higher, but their anti-S1/S2 IgG and anti-RBD IgG antibody levels were interestingly even lower (mean 14.1 S/CO, 8041.7 AU/mL, 81.5 neutr (%), respectively) compared to group 1 who received the same vaccine regimen (three doses of BNT162b2) but were not infected (mean 11.2 S/CO, 8272.2 AU/mL, 98.3 neutr (%), respectively). When individuals with hybrid immunity (two-dose mRNA vaccine regimen + infection) were boosted with BBIBP-CorV, their anti-RBD IgG and anti-S IgA antibody levels were similar, but their anti-S1/S2 IgG level was halved (mean 85.8 neutr (%), 6.9 S/CO, 532.3 AU/mL, respectively) compared to group 2 individuals who received the same vaccine regimen (BNT162b2 + BNT162b2 + BBIBP-CorV) but were not infected between the second and third doses (mean 79 neutr (%), 5,7 S/CO, 1195.8 AU/mL, respectively). The anti-S1/S2 IgG, anti-RBD IgG, and anti-S IgA antibody levels of participants with hybrid immunity before and after the third vaccination are shown in Appendix A, respectively.

## 4. Discussion

Immunogenicity of a third dose of COVID-19 vaccines using a heterologous vaccination strategy is increasingly becoming the focus of the global fight against SARS-CoV-2. To date, only a few studies or clinical trials have been published on heterologous third dose vaccination among healthy individuals [18,19,20], hemato-oncological patients [21], or kidney transplant recipients [22]. Furthermore, all participants of these studies received adenovirus-vector vaccines (ChAdOx1, Oxford-AstraZeneca; Ad26.CoV2.S, Janssen) or mRNA vaccines (BNT162b2, Pfizer-BioNTech; mRNA1273, Moderna) for the first and second doses. To our knowledge, few published articles and preprints [11,13,14,15,16,23,24,25] investigate the immunogenicity of heterologous boosts with a recombinant protein subunit (ZF2001, Anhui Zhifei Longcom, Anhui, China), mRNA vaccine (BNT162b2), or adenovirus-vector vaccine (Convidecia, CanSino Biologicals, Tianjin, China; ChAdOx1; Ad26.COV2.S) after a two-dose inactivated virus (CoronaVac, Sinovac; BBIBP-CorV, Sinopharm) vaccine regimen. Unfortunately, these studies describe only the neutralizing ability, which is undoubtedly key for host protection. However, they do not provide any information about T cell responses, although a heterologous T cell response is critical for SARS-CoV-2 immunity [26,27]. Several publications emphasize that T cell response remains robust even against the Omicron variant, as 93% of CD4 and 97% of CD8 epitopes are 100% conserved across variants, and T cell reactivity is preserved in most vaccinated individuals [28,29,30]. This means that the SARS-CoV-2 Omicron variant partially escapes humoral but not T cell-mediated response in vaccines [31,32,33]; therefore, measuring T cell responses during both homologous and heterologous third dose vaccinations is highly recommended to obtain a comprehensive picture about the state of adaptive immunity.

To address this important knowledge gap, we compared the humoral and T cell-mediated immune responses of healthy adult individuals using homologous or heterologous vaccination strategies after three doses, applying the inactivated virus BBIBP-CorV and/or the mRNA BNT162b2 vaccines in different combinations. All participants received the third dose 6 months following the initial two-dose vaccination. Peripheral blood samples were collected and analyzed on the day of receiving the third dose and 14 days after. In total, three groups were formed according to the vaccine combinations. The first group received a booster dose of BNT162b2 following two doses of BNT162b2; the second received a booster dose of BBIBP-CorV following two doses of BNT162b2, while participants of the third group were boosted with a BNT162b2 vaccine following a two-dose inactivated virus vaccine regimen. A small subset of volunteers from groups 1 and 2 probably underwent a SARS-CoV-2 infection between the second and third immunization, whose antibody and T cell responses were also investigated before and after receiving the third dose.

Six months after the two-dose BNT162b2 or BBIBP-CorV vaccination, both the T cell-mediated and humoral immune responses were radically reduced compared to our previous results measured two weeks after the second vaccination [17]. Comparing the adaptive immunity elicited by two doses of the BNT162b2 and the BBIBP-CorV vaccines after 6 months, the mean level of anti-S1/S2 IgG and neutralizing anti-RBD IgG antibodies was significantly lower in those participants previously immunized with two doses of BBIBP-CorV than in those who received two doses of BNT162b2 (4.75- and 4.3-fold lower, respectively), while the level of anti-S IgA antibody was equally low in both cohorts without significant difference. This is consistent with the results of a recent study showing that 6 months after immunization, the level of neutralization antibodies decreases less after the administration of a two-dose BNT162b2 regimen than after two doses of an inactivated virus vaccine [34]. However, the cumulative number of SARS-CoV-2-specific IFNγ-secreting T cells was almost equal in both cohorts (average 121 SFU/2.5 × 10^5^ PBMC in the BNT162b2 and 118 SFU/2.5 × 10^5^ PBMC in the BBIBP-CorV cohort).

After the third immunization, the anti-SARS-CoV-2 antibody levels were significantly higher in group 1 (three doses of BNT162b2) than in group 2 (BNT162b2 + BNT162b2 + BBIBP-CorV) for anti-S1/S2 IgG, anti-RBD IgG, and anti-S-IgA antibodies, showing that an additional dose of an mRNA vaccine enhances humoral immunity much better than a booster dose of inactivated virus vaccine 6 months after a two-dose mRNA vaccine regimen. This difference is not surprising since we know that three doses of the BNT162b2 vaccine induce an outstanding antibody response and effectively protect individuals against severe COVID-19-related outcomes [35,36]. An even more interesting finding, however, is that there was almost no difference in the antiviral T cell response since the cumulative number of IFNγ-secreting T cells was lower by only 3 SFUs in group 2 (121 vs. 118 SFU/2.5 × 10^5^ PBMC). The most remarkable results were obtained in group 3 (BBIBP-CorV + BBIBP-CorV + BNT162b2), highlighting that the most effective method of using the inactivated virus vaccines is heterologous boosting. Although 2.4-fold lower anti-S1/S2 IgG levels, but similarly high neutralizing anti-RBD IgG and anti-S IgA antibody levels were measured in group 3 participants compared to group 1 participants (immunized with three doses of the BNT162b2 vaccine), the difference between the anti-S1/S2 IgG levels was not statistically significant due to the high interindividual variability that characterized both groups. This is in accordance with the results of a recent study according to which both the BNT162b2 + BNT162b2 + BNT162b2 and the BBIBP-CorV + BBIBP-CorV + BNT162b2 vaccine regimens provided acceptable neutralizing immunity even against Omicron [14]. Surprisingly, an mRNA booster dose 6 months after the two-dose inactivated virus vaccines induced the strongest T cell response (mean 173 SFU/2.5 × 10^5^ PBMC); group 3 participants had an average 43% higher cumulative IFNγ-secreting virus-specific T-cell number compared to virus-naive participants of group 1 immunized with three doses of BNT162b2 (mean 121 SFU/2.5 × 10^5^ PBMC). In group 3, however, we cannot differentiate between virus-naive and virus-experienced individuals, so previous infections must have contributed to the superior T cell response observed in group 3 compared to the virus-naive participants of groups 1 and 2. If virus-experienced individuals were also included in group 1, the average cumulative T cell number in group 3 would be 15% higher than in group 1 (mean 149.8 SFU vs. 172.9 SFU). Interestingly, although individuals of groups 1 and 2 with hybrid immunity (who received two doses of the BNT162b2 and a third dose of the BNT162b2 or BBIBP-CorV vaccines and were also infected with the virus) showed the most intense T cell response (mean 209.8 SFU/2.5 × 10^5^ PBMC), but had a lower level of either neutralizing anti-RBD IgG or anti-S1/S2 IgG antibodies compared to the virus-naive participants of the corresponding groups. This result raises the question of whether the fourth dose of vaccine could significantly increase anti-SARS-CoV-2 antibody levels. To answer this, further studies are recommended concerning the B and T cell-mediated immune responses as well, after a possible fourth vaccination.

In summary, although three doses of the BNT162b2 vaccine induce a strong antibody and T cell response, administration of a booster BNT162b2 vaccine after immunization with a two-dose BBIBP-CorV vaccine regimen results in remarkable antibody response and an even more robust T cell response. Moreover, this T cell response targets not only the spike but also the nucleocapsid, membrane, and envelope proteins, thus providing a more heterogeneous immunity. This is particularly advantageous in light of recent results revealing that N protein is relatively conserved among different SARS-CoV-2 variants and is an excellent immunogen; thus, nucleocapsid vaccine candidates elicit strong protective immunity and robust B cell response [37,38]. A rising number of studies indicate an urgent need for multicomponent vaccines that are not limited to the highly mutable S protein, as new variants of SARS-CoV-2 increasingly escape from immunity targeting the S protein only [39,40].

The main limitations of our work include the small sample size and the lack of presentation of the immune response elicited by three doses of the BBIBP-CorV vaccine, which is a vaccination strategy that has not been used in Hungary.

## 5. Conclusions

Six months after the two-dose immunization against SARS-CoV-2 with the BNT162b2 (Pfizer-BioNTech) or BBIBP-CorV (Sinopharm) vaccine, significant differences were observed regarding the humoral immune responses. Two doses of the BNT162b2 vaccine elicited significantly higher antireceptor-binding domain (anti-RBD) IgG, anti-spike (anti-S1/S2) IgG, and anti-spike (anti-S) IgA levels than two doses of the BBIBP-CorV vaccine. However, the cumulative IFNγ-positive T cell response was almost identical in the two groups. Regarding the adaptive immunity elicited by the third (booster) dose 6 months after the initial vaccine series, three doses of the BNT162b2 vaccine elicited similar anti-RBD IgG, anti-S1/S2 IgG, and anti-S IgA levels compared to the BBIBP-CorV + BBIBP-CorV + BNT162b2 vaccine regimen. However, surprisingly, in the latter case, the cumulative T cell response was 43% higher than after three mRNA shots. The lowest antibody levels were observed in participants who received BNT162b2 + BNT162b2 + BBIBP-CorV vaccine regimens; however, the cumulative T cell response was almost identical to those who received three doses of BNT162b2. Individuals with hybrid immunity who received three vaccine doses (BNT162b2 + BNT162b2 + BBIBP-CorV or BNT162b2 + BNT162b2 + BNT162b2) and were infected with the virus showed an even higher T cell response, but their antibody levels were no longer increased compared to virus-naive participants with the same vaccination series.

## Figures and Tables

**Figure 1 vaccines-10-00539-f001:**
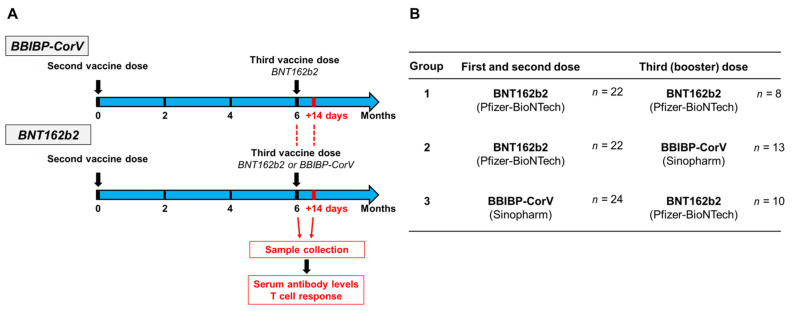
Vaccination schedule and sample collection. Study participants received a third dose of the BBIBP-CorV or BNT162b2 vaccines 6 months after the two-dose vaccine regimen. First samples were collected at this time point. After 14 days of receiving the third dose, peripheral blood samples were obtained, and humoral and T cell-mediated immune responses were quantified (**A**). Participants of the present study were divided into three groups based on the vaccine combinations they received (**B**).

**Figure 2 vaccines-10-00539-f002:**
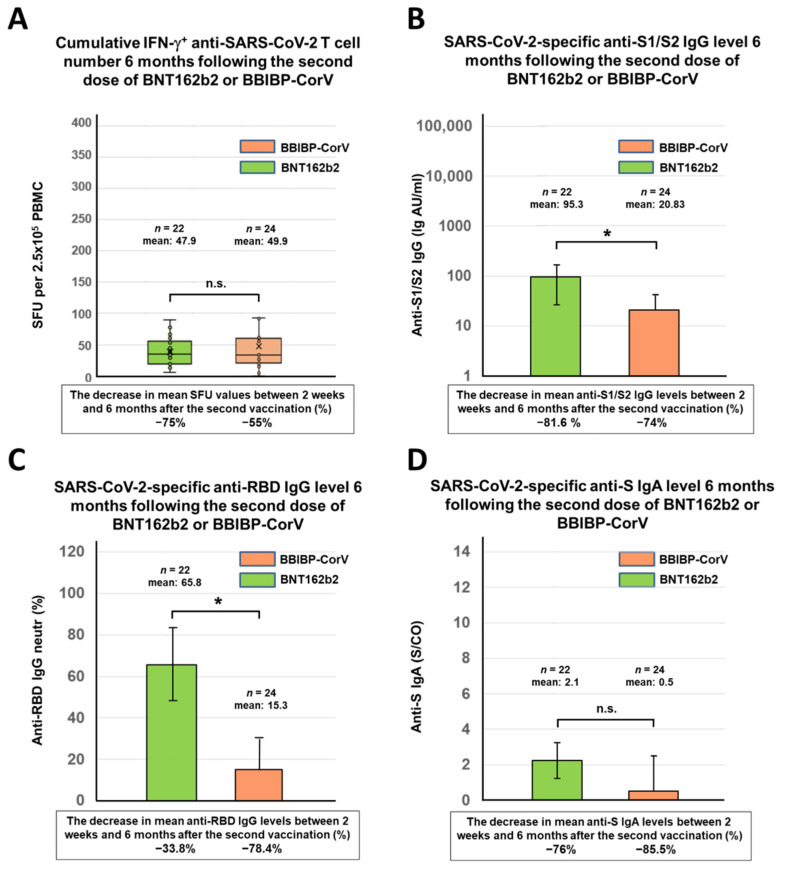
SARS-CoV-2-specific antibody and T cell responses 6 months after the second dose of mRNA (BNT162b2) or inactivated virus (BBIBP-CorV) vaccine. The cumulative IFNγ-positive T cell responses (total SFU against S1, S2, N, M, and E antigens) were evaluated by the T-SPOT Discovery SARS-CoV-2 ELISpot assay for 22 virus-naive participants 6 months after receiving two doses of the BNT162b2 vaccine and for 24 individuals 6 months after receiving the second dose of the BBIBP-CorV vaccine (**A**). The SARS-CoV-2-specific anti-S1/S2 IgG levels (**B**), the anti-RBD IgG levels (**C**), and the anti-S IgA levels (**D**) of the total of 46 participants were determined by the LIAISON SARS-CoV-2 S1/S2 IgG test, the SARS-CoV-2 surrogate virus neutralization test (sVNT), and the SARS-CoV-2 anti-S IgA assay, respectively. The values of bar graphs are presented as mean ± standard deviation. Box plots display the median values with the interquartile range (lower and upper hinge) and ±1.5 fold the interquartile range from the first and third quartile (lower and upper whiskers). Statistical analyses were performed applying Student’s *t*-test, and *p*-values of <0.05 were considered statistically significant (*) while *p*-values of >0.05 were considered non-significant (n.s.).

**Figure 3 vaccines-10-00539-f003:**
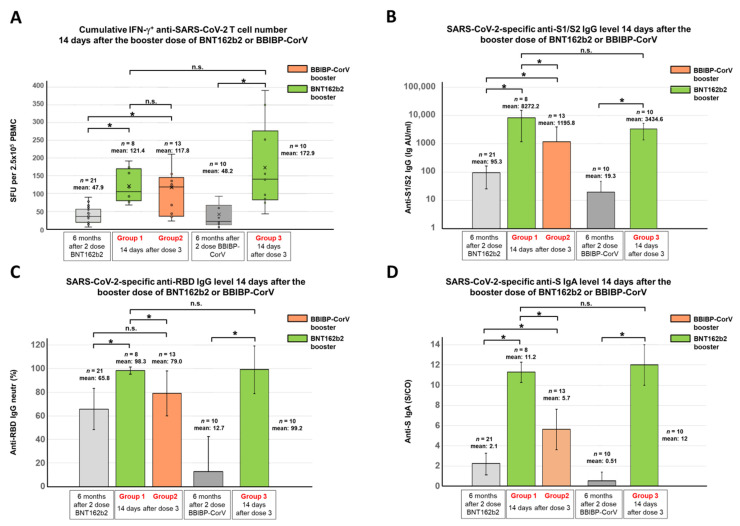
SARS-CoV-2-specific antibody and T cell responses before and after the third (booster) dose of mRNA (BNT162b2) or inactivated virus (BBIBP-CorV) vaccine. The cumulative IFNγ-positive T cell responses (total SFU against S1, S2, N, M, and E antigens) were evaluated by the T-SPOT Discovery SARS-CoV-2 ELISpot assay for 21 individuals who were immunized with two doses of the BNT162b2 vaccine and boosted with the BNT162b2 (*n* = 8) or BBIBP-CorV (*n* = 13) vaccines and moreover for 10 individuals who were immunized with two doses of the BBIBP-CorV and boosted with the BNT162b2 vaccine (**A**). The SARS-CoV-2-specific anti-S1/S2 IgG levels (**B**), the anti-RBD IgG levels (**C**), and the anti-S IgA levels (**D**) of the total of 31 participants were determined by the LIAISON SARS-CoV-2 S1/S2 IgG test, the SARS-CoV-2 surrogate virus neutralization test (sVNT), and the SARS-CoV-2 anti-S IgA assay, respectively. Blood samples were obtained 14 days after receiving the booster dose of the BNT162b2 or BBIBP-CorV vaccine. The gray bars represent the mean cumulative SFU, anti-S1/S2 IgG, anti-RBD IgG, and anti-S IgA levels 6 months after the second vaccination. Statistical analyses were performed applying the Wilcoxon signed-rank test or Student’s *t*-test as appropriate, and *p*-values of <0.05 were considered statistically significant (*) while *p*-values of >0.05 were considered non-significant (n.s.). The values of bar graphs are presented as mean ± standard deviation. Box plots display the median values with the interquartile range (lower and upper hinge) and ±1.5 fold the interquartile range from the first and third quartile (lower and upper whiskers).

## Data Availability

All data, materials, and methods used in the analysis are available from the corresponding author by request.

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
