# Peer review of "Antibody and T Cell Responses against SARS-CoV-2 Elicited by the Third Dose of BBIBP-CorV (Sinopharm) and BNT162b2 (Pfizer-BioNTech) Vaccines Using a Homologous or Heterologous Booster Vaccination Strategy"

_vaccines, 2022, doi:10.3390/vaccines10040539_

Round 1

Reviewer 1 Report

In the paper the authors set out to compare the humoral and T cell mediated immune responses against SARS-CoV-2 elicited by the BNT162b2 mRNA-vaccine following two doses of the BBIBP-CorV in-activated virus vaccine (heterologous vaccination strategy) and compared them to a homologous vaccination strategy using three doses of the BNT162b2 vaccine. A total of 52 healthy individuals were investigated regarding the humoral and the T cell-mediated immune responses using serological assays and T cell ELISpot assay with groups sizes ranging from 8-24 individuals. The authors find that the homologous vaccine strategy with BNT162b2 induced stronger humoral immune responses, while the T cell-mediated immune responses were similar.

The experimental design and methodology appear sound, although the group sizes could always be larger. The way the data is reported could, however, be improved.

Major comments:

The figures are not ordered according to the order they are mentioned in the text (fig 1 then 3A then 2A,B).

A few individuals were deemed to probably have experienced a mild SARS-CoV-2 infection after receiving the second dose of the vaccine, based on > 8 SFU/2.5 × 105 PBMC for N and M antigens. How was this threshold selected? Is there evidence in the literature to support this threshold?

It is unclear why T cell-response data (fig 2A and B) are presented as boxplots, whereas humoral response data (fig 2C-H) are presented as bar graphs?

It is unclear what the error bars in Fig 2C-H represent?

It should be mentioned what the gray bars are in fig 2C-H.

The paired nature of the data and choice of statistical method (Wilcoxon signed rank test) could be highlighted by connecting paired data with lines.

Abstract, discussion and Row 240: Statistical comparisons are made between Group 1 and 3, but the statistical test used is unclear and the comparison is not evident from the figures. This result is highlighted in both the abstract and the discussion “The most remarkable result… (row 329)” and deserves a figure of its own if this is the main take home message of the paper. The clinical implications of this data if they are replicated and confirmed could be discussed. Would there be an advantage starting with an inactivated virus-vaccine that is boosted with an mRNA-vaccine over other vaccine strategies?

Minor comments:

The figure legends could be shortened by removing some methodological details.

Row 236: remove “greatly”.

Row 136: should be “… twice as high in the …”

Author Response

First of all, we thank your careful work and greatly appreciate your comments and suggestions. Based on these comments, we carried out a careful revision of the manuscript. We believe we answered all the concerns, with special attention to improve writing and the presentation of our results. We used MDPI’s English editing service thus our manuscript was checked by a native English-speaking colleague (English Editing ID: english-41669).  All the modifications of the editor (298 in total) were accepted but our modifications are still visible by the “Track change” function.  

Reviewer 1.

In the paper the authors set out to compare the humoral and T cell mediated immune responses against SARS-CoV-2 elicited by the BNT162b2 mRNA-vaccine following two doses of the BBIBP-CorV in-activated virus vaccine (heterologous vaccination strategy) and compared them to a homologous vaccination strategy using three doses of the BNT162b2 vaccine. A total of 52 healthy individuals were investigated regarding the humoral and the T cell-mediated immune responses using serological assays and T cell ELISpot assay with groups sizes ranging from 8-24 individuals. The authors find that the homologous vaccine strategy with BNT162b2 induced stronger humoral immune responses, while the T cell-mediated immune responses were similar.

The experimental design and methodology appear sound, although the group sizes could always be larger. The way the data is reported could, however, be improved.

Major comments:

  1. The figures are not ordered according to the order they are mentioned in the text (fig 1 then 3A then 2A, B).

We made two separate figures (Figure 2 and Figure 3) from the previous Figure 2. The new Figure 2 shows only the 6-month results comparing the T cell-mediated and antibody response elicited by the two-dose BBIBP-CorV or BNT162b2 vaccine. These results are discussed in a separate subsection within the results section and now have a separate Figure as well. The new Figure 3 shows our results after the third vaccination, comparing the T cell-mediated and humoral immune responses before and after receiving the third vaccine dose. The original Figure 3 was relegated to supplementary material as suggested by the other reviewer and was removed from the manuscript. Thus, Figures 1, 2, and 3 are now discussed in the text in the appropriate order.

  1. A few individuals were deemed to probably have experienced a mild SARS-CoV-2 infection after receiving the second dose of the vaccine, based on > 8 SFU/2.5 × 105 PBMC for N and M antigens. How was this threshold selected? Is there evidence in the literature to support this threshold?

The manufacturer (Oxford Immunotec) recommends this threshold in the case of T-Spot.Covid kit: “The test result is Reactive if (Panel A-Nil) and/or (Panel B-Nil) ≥ 8 spots.” Therefore, we applied the same criterion. The manufacturer's description is available at the following link: https://www.tspotcovid.com/wp-content/uploads/sites/5/2022/03/PI-T-SPOT.COVID-IVD-UK-v4.pdf

In the case of T-SPOT.TB kit from Oxford Immunotec, the positivity of the result is recommended also from 8 spots: “The test result is Positive if (Panel A-Nil) and/or (Panel B-Nil) ≥ 8 spots.” Manufacturer's description:

https://www.oxfordimmunotec.com/international/wp-content/uploads/sites/3/Final-File-PI-TB-US-V6.pdf

  1. It is unclear why T cell-response data (fig 2A and B) are presented as boxplots, whereas humoral response data (fig 2C-H) are presented as bar graphs?

There is a technical reason for this: a boxplot cannot be represented on a logarithmic scale due to the nature of the graph type, data can only be represented using a linear ordinate. There were huge differences between the anti-S1/S2 IgG antibody levels (e.g., 20.83 AU/ml vs. 8272.2 AU/ml), thus when plotted with a linear ordinate, the low antibody values were not readable from the axis, and the difference between the low antibody levels was not visible. Therefore, we chose the logarithmic ordinate and the bar graphs in the case of anti-S1/S2 IgG levels. For uniformity, we decided to keep the boxplots for T cell-mediated responses and show the antibody levels uniformly using bar graphs.

  1. It is unclear what the error bars in Fig 2C-H represent?

The values are presented as mean ± standard deviation. We highlighted in figure legends: „The values of bar graphs are presented as mean ± standard deviation.” We also indicated the parameters of the boxplots in figure legends: „Box plots display the median values with the interquartile range (lower and upper hinge) and ± 1.5 fold the interquartile range from the first and third quartile (lower and upper whiskers).”

  1. It should be mentioned what the gray bars are in fig 2C-H.

The new Figure 3 has been thoroughly redesigned. Below the X-axis, it is now clearly indicated what the light-gray (6-month results after two BNT162b2 vaccinations) and dark-gray columns (6-month results after two BBIBP-CorV vaccinations) represent.

  1. The paired nature of the data and choice of statistical method (Wilcoxon signed rank test) could be highlighted by connecting paired data with lines.

The new Figure 3 has been thoroughly redesigned. In the case of new Figure 3, the paired data is now connected by lines in each case.

  1. Abstract, discussion and Row 240: Statistical comparisons are made between Group 1 and 3, but the statistical test used is unclear and the comparison is not evident from the figures. This result is highlighted in both the abstract and the discussion “The most remarkable result… (row 329)” and deserves a figure of its own if this is the main take home message of the paper. The clinical implications of this data if they are replicated and confirmed could be discussed. Would there be an advantage starting with an inactivated virus-vaccine that is boosted with an mRNA-vaccine over other vaccine strategies?

The new Figure 3 shows the results of groups 1, 2, and 3 together now, so the previous panels 2A and 2B were merged, as well as the other panel pairs (2C-2D, 2E-2F, 2G-2H). Thus, the results of groups 1 and 3 can now be compared, and their statistical comparison is also indicated in the figure. The main limitation of our work is the small sample size, but we are expanding the number of participants, and we are constantly monitoring the clinical status of the participants. If sufficient time elapses, the clinical implications of this heterologous vaccination strategy will be discussed. In answer to the last question, it probably has an advantage starting with an inactivated virus vaccine and then boosting with an mRNA-vaccine over other vaccine strategies, because the inactivated virus vaccine induces immune memory against other viral proteins in addition to the spike protein. When the spike protein has many mutations, the efficiency of protection may decrease not as great as e.g. for an mRNA vaccine encoding only the spike protein. However, we do not have enough data yet to confidently state this therefore we are constantly monitoring the clinical status of these participants. On the other hand, based on our results, it is certainly recommended that the huge number of people worldwide who have been immunized with two doses of inactivated virus vaccine should receive an mRNA vaccine as soon as possible to gain a significant T cell memory.

Minor comments:

  1. The figure legends could be shortened by removing some methodological details.

As we made two separate figures instead of the previous Figure 2, the figure legends are shortened and much more understandable.

  1. Row 236: remove “greatly”.

“greatly” has been removed.

  1. Row 136: should be “… twice as high in the …”

We corrected.

Reviewer 2 Report

The authors have presented some important data on the use of heterologous boosts for people receiving an inactivated COVID-19 vaccine. However, the presentation of results is very confusing. A small sample size seems to have been split up into multiple groups with a large number of comparisons across multiple timepoints. Figures are generally unhelpful with multiple comparisons across figure panels. The authors need to simplify their grouping system, decide exactly what they want to compare, and present their data neatly. Additional data (e.g. for those with prior infection, which seems unconfirmed and small sample size) should be relegated to supplementary material. 

  1. For efficacy of CoronaVac, should really cite the PROFISCOV study instead of the Lancet study by Tanriover et al which is only an interim analysis. The PROFISCOV study results are available as a preprint in the SSRN server and concludes a primary efficacy of ~ 50% against symptomatic COVID.
  2. Figure 1B: there are 5 rows but only 3 assigned groups, which is confusing. Even the legend says patients are divided into five groups, but only 3 of these seem to be formally designated. 
  3. Figure 3A is being cited in the text before figure 2. Please rearrange in order to cite figures in correct order. 
  4. I can't understand why data in figures 2A and B cant be shown in the same panel? Same goes for all figure panel pairs in figure 2. It is really confusing to interpret as it stands now. Line 176 - 183 confirms that statistical comparisons are being made across panels. Please rethink how the figures could be condensed together for easier interpretation. 
  5. The group 1/2/3 designation used in figure 1B is not used in subsequent figures adding to the confusion.
  6. Figure 1B seems to suggest that not all individuals who received BNT/Sinopharm vaccine actually received the third dose?? Really need to clarify why the non-boosted individuals are included in this study. 

Author Response

First of all, we thank your careful work and greatly appreciate your comments and suggestions. Based on these comments, we carried out a careful revision of the manuscript. We believe we answered all the concerns, with special attention to improve writing and the presentation of our results. We used MDPI’s English editing service thus our manuscript was checked by a native English-speaking colleague (English Editing ID: english-41669).  All the modifications of the editor (298 in total) were accepted but our modifications are still visible by the “Track change” function. 

Reviewer 2.

The authors have presented some important data on the use of heterologous boosts for people receiving an inactivated COVID-19 vaccine. However, the presentation of results is very confusing. A small sample size seems to have been split up into multiple groups with a large number of comparisons across multiple timepoints. Figures are generally unhelpful with multiple comparisons across figure panels. The authors need to simplify their grouping system, decide exactly what they want to compare, and present their data neatly. Additional data (e.g. for those with prior infection, which seems unconfirmed and small sample size) should be relegated to supplementary material. 

Two separate figures were created (new Figure 2 and Figure 3) from previous Figure 2, and the original Figure 3 was relegated to supplementary material as you suggested. Participants were divided into 3 groups, and these groups do not include individuals with hybrid immunity thus Figure 1B now contains only 3 rows. Although the results of participants with hybrid immunity are discussed in the main text, their data are not included in either the new Figures 2 or 3, only in the supplementary material. The new Figure 2 shows only the 6-month results, while the new Figure 3 shows the results of groups 1, 2, and 3 together before and after the third vaccination. The previous panels 2A and 2B were merged, as well as the other panel pairs (2C-2D, 2E-2F, and 2G-2H), thus the results of groups 1, 2, and 3 can now be compared, and their statistical comparison is also shown in Figure 3.

  1. For efficacy of CoronaVac, should really cite the PROFISCOV study instead of the Lancet study by Tanriover et al which is only an interim analysis. The PROFISCOV study results are available as a preprint in the SSRN server and concludes a primary efficacy of ~ 50% against symptomatic COVID.

Reference # 5 was replaced with the reference of the PROFISCOV study, and the primary efficacy of CoronaVac was corrected to 50.8% in the main text.

  1. Figure 1B: there are 5 rows but only 3 assigned groups, which is confusing. Even the legend says patients are divided into five groups, but only 3 of these seem to be formally designated

There are 3 rows in Figure 1B now, with 3 group designations, and the number of groups was corrected to 3 in the figure legend as well.

  1. Figure 3A is being cited in the text before figure 2. Please rearrange in order to cite figures in correct order.

We created two separate figures (new Figure 2 and Figure 3) from the previous Figure 2, and the original Figure 3 was relegated to supplementary material as you suggested. The new Figure 2 shows only the 6-month results, while the new Figure 3 shows the results of groups 1, 2, and 3 together before and after the third vaccination. Therefore, the figures are now referred to in the appropriate order in the text.

  1. I can't understand why data in figures 2A and B cant be shown in the same panel? Same goes for all figure panel pairs in figure 2. It is really confusing to interpret as it stands now. Line 176 - 183 confirms that statistical comparisons are being made across panels. Please rethink how the figures could be condensed together for easier interpretation. 

We fully agree with your proposal, so we fixed the mistake. The previous panels 2A and 2B were merged, as well as the other panel pairs (2C-2D, 2E-2F, and 2G-2H), thus the results of groups 1, 2, and 3 can now be compared, and their statistical comparison is also shown in Figure 3.

  1. The group 1/2/3 designation used in figure 1B is not used in subsequent figures adding to the confusion.

The group 1/2/3 designation used in figure 1B is now used in Figure 3 and indicated below the abscissa.

  1. Figure 1B seems to suggest that not all individuals who received BNT/Sinopharm vaccine actually received the third dose?? Really need to clarify why the non-boosted individuals are included in this study. 

An important part of the study was to determine the magnitude of T cell-mediated and humoral immune response 6 months after the two-dose BNT162b2 or BBIBP-CorV vaccination. A total of 22 participants from the BNT162b2 cohort and 24 participants from the BBIBP-CorV cohort returned for an additional investigation 6 months after receiving the two-dose vaccine regimen. Their results are now summarized in a separate figure (new Figure 2). The new Figure 2 also shows how much their immunological parameters decreased compared to the results measured two weeks after the second vaccination. These results are now discussed also in the introduction, results, and discussion sections in the revised manuscript because these results were not highlighted in these sections of the original manuscript, just in the abstract section. In our view, it is not appropriate to remove the data of those participants who did not receive the third dose of the vaccines from subsection 3.1 of the results and Figure 2, because these are very important data for the 6-month follow-up.

Another important part of the study was the determination of the T cell-mediated and humoral immune response 14 days following the third vaccination. These results are now also summarized in a separate figure (new Figure 3) in the revised manuscript. An important change, however, is that Figure 3 shows only the 6-month results of those individuals who also received the third dose of the vaccines. Data of individuals who did not receive the third dose were removed from all panels, therefore new bar graphs and boxplots (gray bars and boxplots) were generated for Figure 3 indicating the 6-month results. You can see that for group 3 we show the results of the same 10 participants before and after the third vaccination. Similarly, the 6-month results of 1 participant from group 1 were omitted because he did not receive the third vaccine dose. In our opinion, comparing these results in this way is much more appropriate.

Round 2

Reviewer 2 Report

  1. Again a mistake in Figure 1B. Group 3 shows 3 doses of Sinopharm in the table, but the discussion and section 3.2 indicates that these Sinopharm X 2 patients actually received a heterologous boost with BNT162b2. 
  2. Please comment on whether past infections might have accounted for the superior T cell responses observed in the Sinopharm X 2 + BNT162b2 group. 
  3. No Sinopharm X 3 group should be noted as a limitation.  

Author Response

Dear Editor,

We are grateful for the opportunity to improve and resubmit our manuscript. We appreciate the time and effort that the Reviewers have dedicated to providing valuable feedback on our manuscript, and we are grateful for their constructive criticism.

The specific changes made in response to the reviewer comments are detailed below (all modifications are visible by the “Track change” function in the manuscript):  

Reviewer 2.

  1. Again a mistake in Figure 1B. Group 3 shows 3 doses of Sinopharm in the table, but the discussion and section 3.2 indicates that these Sinopharm X 2 patients actually received a heterologous boost with BNT162b2. 

Thank you for your remark, we have improved the third row of Figure 1B, which details the vaccination strategy of Group 3. Group 3 participants received a BNT162b2 booster, not Sinopharm. The manuscript now contains the corrected figure (page 4).

  1. Please comment on whether past infections might have accounted for the superior T cell responses observed in the Sinopharm X 2 + BNT162b2 group. 

As requested, we expanded the section of the Discussion and noted that past infections must have affected the T cell response of group 3 participants (page 9):

„In group 3, however, we cannot differentiate between virus-naive and virus-experienced individuals, so previous infections must have contributed to the superior T cell response observed in group 3 compared to the virus-naive participants of groups 1 and 2. If virus-experienced individuals were also included in group 1, the average cumulative T cell number in group 3 would be 15% higher than in group 1 (mean 149,8 SFU vs. 172.9 SFU).”

  1. No Sinopharm X 3 group should be noted as a limitation.  

As requested, we expanded the section of the Discussion and indicated the main limitations of the study (page 9):

„The main limitations of our work include the small sample size and the lack of presentation of the immune response elicited by three doses of the BBIBP-CorV vaccine, which vaccination strategy has not been used in Hungary. ”
